# Effectiveness of health partners coordination for COVID-19 pandemic response in Nepal

**Sangeeta Kaushal Mishra[1], Samir Kumar Adhikari[1], Pavan Kumar Sah[1], Allison Eugenio Gocotano[2], Subash Neupane[2], Barsha Thapa[2], Gaurav Devkota[2]***

1 Ministry of Health and Population, Government of Nepal, Kathmandu, Nepal, 2 World Health Organization Country Office for Nepal, Lalitpur, Nepal

* devkotag@who.int, gdevkota74@gmail.com

## Abstract

### Background

Nepal established the health partner coordination for COVID-19 response based on national and international plans and framework to support information-sharing for decision- making and course correction. This paper aims to assess the performance of COVID-19 Health Partner Coordination response in Nepal by adopting coordination best practices and tools from the international humanitarian cluster system.

### Methods

Secondary data from unpublished documents on health partner coordination for COVID-19 response in Nepal was collected and analysed from April 2023 to May 2023. The secondary data were from the review of the health partner coordination meetings conducted for COVID-19 response using a researcher-developed document review tool, responses of health partners on a survey tool adapted from the cluster coordination performance monitoring tool from the Global Health Cluster, and transcript of Focused Group Discussion among health partners. Descriptive analysis of quantitative information and thematic analysis with predefined themes of qualitative information were performed using MS Excel and MS Word respectively. A written approval from the Ministry of Health and Population and an ethical clearance from the Nepal Health Research Council was obtained before conducting the study.

### Results

More than three-fifths of the meetings showed good results in conduction, process, participation, and documentation of meeting action points with improvement required for follow-up (22.2%). Assessment of health partner coordination subfunctions resulted in either 'good' (>75.0%) or 'satisfactory' (50.1%-75.0%) except for prioritization based on analyses, which was 'unsatisfactory' (<50%). Partners admired good practices of health partner coordination, pointed out some issues, and provided recommendations.

**Data Availability Statement:** All relevant data are within the article and its supporting information files.

**Funding:** The author(s) received no specific funding for this work.

**Competing interests:** The authors have declared that no competing interests exist.

## Conclusion

COVID-19 has highlighted the importance of effective coordination of health sector for response to a pandemic. Continuation of this practice after addressing the areas for improvement will contribute to health preparedness and readiness for future disasters or public health emergencies.

## Introduction

World Health Organization (WHO) declared COVID-19 as Public Health Emergency of International Concern (PHEIC) on 30 January 2020 and later as a pandemic on 11 March 2020 [1]. Nepal confirmed its first COVID-19 case on 23 January 2020 [2] and promptly implemented various health interventions like public health measures but still needed stronger coordination to prevent and control COVID-19 pandemic [3]. As stated in United Nations General Assembly resolution 46/182 of 1991 (paragraphs 4–6), "national authorities have the primary responsibility for taking care of the victims of the natural disasters and other emergencies occurring in their territories" [4], focus of Nepal's national authorities was on protecting health and saving lives of people while minimizing the disruptive effect of the pandemic on the economy and society, which called for multisectoral coordination. This is reiterated in the purpose and scope of the International Health Regulations (IHR) which are "*to prevent, protect against, control and provide a public health response to the international spread of disease in ways that are commensurate with and restricted to public health risks, and which avoid unnecessary interference with international traffic and trade* [5]".

Coordination can be defined as "the attitudes, behaviors, and outcomes of joint determination of common (interorganizational relationship) goals" [6]. However, coordination during humanitarian response to a disaster or public health emergency is not as straightforward as a set of specific behaviors but is rather the first step towards an effective, efficient, and sustainable response that builds the foundations of short- and long-term recovery [7]. The Humanitarian Cluster System led by United Nations Office for the Coordination of Humanitarian Affairs (UN OCHA) is a coordination mechanism to improve response coordination between humanitarian organizations (UN and non-UN) working in the main sectors of response (e.g., health, nutrition, shelter, etc.) with systematic and standardized procedures among all sectors, including systems for cross-sector communication to ensure all populations are reached, and to avoid duplication during a humanitarian response [8].

To ensure accountability to the affected populations, the Inter-Agency Standing Committee (IASC) endorsed the Cluster Coordination Performance Monitoring (CCPM) process as a standardized monitoring system to review the performance of country-level cluster coordination in six core cluster functions (supporting service delivery, informing strategic decisions of the Humanitarian Coordinator/Humanitarian Country Team, planning and strategy development, advocacy, monitoring and reporting, contingency planning and preparedness) along with accountability to the affected population [9]. As the Humanitarian Cluster Lead for health, and the leading organization for COVID-19 response, WHO proposed ten pillars in the COVID-19 Strategic Preparedness and Response Plan. The first pillar focused on coordination, planning, financing, and monitoring to ensure coherence and operational alignment of the response at national and subnational levels. Moreover, the coordination pillar is expected to serve as the platform for ongoing decision-making and course correction based on available information [10].

In Nepal, the humanitarian cluster approach was adapted in the National Disaster Response Framework (2013) where it specifies that the lead government agency for the Health Pillar is the Ministry of Health and Population (MoHP) with WHO as co-lead agency [11]. The same approach was implemented during the COVID-19 pandemic for health partner coordination wherein the Health Coordination Division (HCD) of the MoHP acted as lead and WHO Nepal as co-lead. National and subnational coordination was conducted through, and with information management and secretariat support by, the Health Emergency Operation Center (HEOC) at national and provincial levels. The HCD chaired national-level coordination meetings among health partners.

The first coordination meeting among health partners was held on 9 April 2020 [2], and all of the meetings were chaired by MoHP and co-chaired by WHO. The meetings included information-sharing presentations from government and partners to discuss the operational landscape, review the gaps, discuss next steps and agree on priority action points. To respect public health and social measures, all meetings were conducted virtually with participation from national and subnational levels. Decisions and action points were shared with all the partners and followed-up through the responsible focal points to demonstrate transparency and shared accountability.

The HCD led the coordination of all the health partners for COVID-19 response following the provisions from the National Disaster Response Framework. These coordination meetings also provided a platform to share information and experiences, discuss challenges, identify gaps, and support coordinated actions. As the demand for the COVID-19 response diminished, HCD with support from WHO reviewed the effectiveness of the health partner coordination meetings to explore the usefulness of the coordination platform and identify areas for improvements. This paper aims to assess the performance of the Health Partner Coordination established for COVID-19 response in Nepal.

## Methods

This study was conducted between April 2023 and May 2023 based on the secondary information available at the Ministry of Health and Population on the health partner coordination for COVID-19 response. This includes health partner coordination meetings, health partner coordination performance monitoring survey, and focus group discussion with selected health partners. Collection of data and information from different sources on the effectiveness of the health partner coordination for COVID-19 response has ensured data triangulation in the study [12].

### Data collection and tools

First, the available unpublished documents and files of the meetings (agenda, presentations, and minutes) conducted by HCD with health partners for COVID-19 response from March 2020 to February 2022 were reviewed using a tool (S1 File) that was developed from the meeting evaluation checklist [13] and post-meeting assessment steps [14]. The tool assessed the meetings based on four main themes of conduction, process, participation, and follow-up actions/next steps.

Second was the information available from the responses by the health partners to the survey conducted by MoHP using a survey tool modified from the cluster coordination performance monitoring tool [15] from the Global Health Cluster (S2 File). This tool was pretested among the Provincial Health Emergency Operation Centers before it was disseminated to the health partners for response. The survey was administered from 30 May 2022 to 13 June 2022 by email to the primary focal point of each of the 82 organizations who had participated at least once in the health coordination meetings.

Third was the information available from the Focus Group Discussion (FGD) transcript conducted by HCD among focal person of two UN agencies, two donors, two INGOs and one NGO. These stakeholders were selected based on their participation in more than 50% of the last 20 health coordination meetings conducted for COVID-19 response, with documented attendance list. The FGD was conducted using an interview guideline (S1 Table) with questions to discuss on the major themes as conduction, process, participation, and way forward. The available transcript analysed is the English translation of the recorded FGD which was conducted in Nepali.

## Ethical clearance

A written approval from the Ministry of Health and Population was obtained to collect and analyze these unpublished documents and files. Ethical clearance from Nepal Health Research Council (8/2023 P) was obtained before conducting this analysis. As the methodology included collection and analysis of secondary information, no human subjects were involved in the study. However, deidentified data were used for analysis and codes were provided to maintain anonymity of the person who had provided information (S1 Data).

## Data analysis

Descriptive analysis was performed using MS Excel on the data that were available from the document review tool. The available responses on the survey tool were analysed using four categories, i.e., an overall performance score of greater than 75.0% defined to be good, 50.1%-75.0% as satisfactory, 25.1%-50.0% as unsatisfactory, and less than or equal to 25.0% as weak. This was applied to each of the six core functions and with the addition of accountability to the affected population. Thematic analysis was performed in MS Word on the available transcript of the FGD.

Anonymity of personnel and organization was maintained during analysis of data and results description. Codes were provided to the FGD participants which were used during analysis. Privacy of all the data and information is observed through a password protected data repository managed by the Health Emergency Operation Center, MoHP. This data repository is to be maintained for a period of ten years.

## Results

The findings from the analysis of unpublished documents and files of the meetings conducted by health coordination division with health partners for COVID-19 response, analysis of the information available through a survey tool modified from the cluster coordination performance monitoring tool, and the analysis of the information available as transcript of a Focus Group Discussion (FGD) are as below:

## Health coordination meetings' document review

A total of 72 meetings were conducted till mid- February 2022. The frequency of the meetings varied as per the COVID-19 situation, that is from weekly, two-weekly, monthly, to as required (Fig 1). Each meeting had aproximatley 55 individual participants. On average, 43.5 minutes were spent on presentations and 11.5 minutes for discussions in a one-hour meeting. The country level coordination, planning and monitoring pillar had the longest time allocated (average of 17.2 minutes), followed by surveillance, rapid response teams, and case investigation pillar (average of 6.3 minutes).

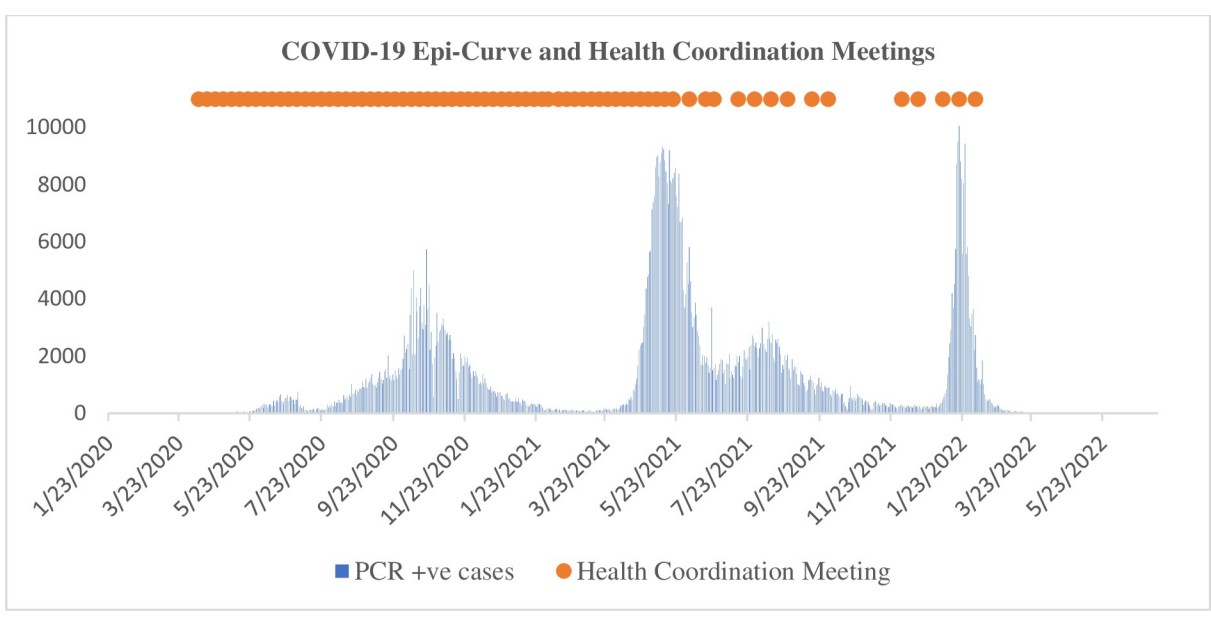

**Fig 1. COVID-19 epidemic curve of Nepal and health coordination meetings, Jan 2020 to June 2022.**

All the agenda items were covered in 80.6% of the meetings while only 66.7% of the meetings had follow-up action points to review from the preceding meeting. Partners presented in 63.9% of the meetings with at least half of the invitees present in 73.6% of the meetings and at least half of the concerns addressed/recognized/acknowledged in 66.7% of the meetings. Finally, follow-up actions and next steps were fully documented at 100.0% of meeting while there were room for improvement in action point completion i.e., "who will be doing what activities and when" was only documented in 22.2% of the meetings (Table 1).

## Performance monitoring of health partner coordination through survey

Out of the 82 survey recipients, responses were received from 26 (31.7%) organizations. While the response rate was low, the organizations which responded had participation rate to the

**Table 1. Analysis of health coordination meeting documents.**

| Themes | Questions | | Number | Percentage |
|---|---|---|---|---|
| Conduction | Was the purpose of the meeting met i.e., all the agenda items covered? | No | 14 | 19.4 |
| | | Yes | 58 | 80.6 |
| | Were the action points of previous meeting followed up? | No | 24 | 33.3 |
| | | Yes | 48 | 66.7 |
| Process | Were presentations from partners included in the meeting? | No | 26 | 36.1 |
| | | Yes | 46 | 63.9 |
| Participation | Did all or most (more than 50%) invitees attend? | No | 19 | 26.4 |
| | | Yes | 53 | 73.6 |
| | Was the participation as expected and as required? (addressed/recognized/acknowledged at least 50% queries) | No | 24 | 33.3 |
| | | Yes | 48 | 66.7 |
| Follow-up actions and next steps | Were the next steps and action items fully documented at the end of the meeting? | No | 0 | 0.0 |
| | | Yes | 72 | 100.0 |
| | Were at least one action point complete (i.e., include who does what and when)? | No | 56 | 77.8 |
| | | Yes | 16 | 22.2 |

**Table 2. Analysis of the responses in the adapted cluster coordination performance monitoring tool.**

| S. N. | Core functions and subfunctions | Results |
|---|---|---|
| **1** | **Supporting service delivery** | |
| 1.1 | Coordinating to ensure that service delivery is driven by the agreed strategic priorities | *Good* |
| 1.2 | Developing mechanisms to eliminate duplication of service delivery | *Satisfactory* |
| **2** | **Informing strategic decision-making of the Humanitarian Coordinator (HC)/ Humanitarian Country Team (HCT) for the humanitarian response** | |
| 2.1 | Needs assessment and gap analysis | *Satisfactory* |
| 2.2 | Analysis to identify (emerging) gaps, obstacles, duplication, and cross-cutting issues | *Satisfactory* |
| 2.3 | Prioritization grounded in response analysis | *Unsatisfactory* |
| **3** | **Planning and strategy development** | |
| 3.1 | Developing sectoral plans, objectives, indicators directly supporting HC/HCT strategic priorities | *Satisfactory* |
| 3.2 | Application and adherence to existing standards and guidelines | *Good* |
| 3.3 | Clarifying funding needs, prioritization, and cluster contributions to HC funding considerations | *Satisfactory* |
| **4** | **Advocacy** | |
| 4.1 | Identifying advocacy concerns to contribute to HC and HCT messaging and action | *Satisfactory* |
| 4.2 | Undertaking advocacy activities on behalf of cluster participants and the affected population | *Satisfactory* |
| **5** | **Monitoring and reporting on the implementation of the cluster strategy and results** | *Satisfactory* |
| **6** | **Preparedness for recurrent disasters whenever feasible and relevant** | *Satisfactory* |
| **7** | **Accountability to affected populations** | *Satisfactory* |

meetings of more than 50.0% whereas the organizations which did not respond had participation rate to the meetings of less than 50.0% (out of the 20 meetings with information available on participants' details).

Two subfunctions yielded good results by scoring more than 75.0% score (Table 2). First is the "Coordinating to ensure that service delivery is driven by the agreed strategic priorities" subfunction of the supporting service delivery function which covers activities related to coordination meetings like regularity, participation, partners' decision-making power and ability to follow-up decision made, usefulness in discussing needs, gaps, and priorities, and agreement on strategic direction. Second is the "Application and adherence to existing standards and guidelines" subfunction of the planning and strategy development function which covers the activities like agreement on the technical standards and guidance by cluster members and its application by the responding organization. "Prioritization grounded in response analysis" subfunction of the informing strategic decision-making function yielded 'Unsatisfactory' result, i.e., 25.0%-50.0% score. This covers the activities on the response planning and prioritization, supported by joint analyses.

## Thematic analysis of the focus group discussion

**Conduction.** Participants responded to the HCD's initiation of the health coordination meeting as a sign of good leadership and deemed them relevant in the context of COVID-19 pandemic response. Also, the fixed day and time for the meeting and the sharing of relevant documents were highly applauded.

"...the excitement, information sharing, has helped activate cluster rapidly. The leadership was appreciative...Next, fixed date and time was set, this somehow helped in time

*management and fixed schedule was maintained throughout the period. . . Reference documents, guidelines were shared time and again during and after the meeting"* -

Participant F

*"In my opinion, the information flow was relevant. Especially, during the second wave it worked as a platform to obtain relevant information through one door mechanism."*—Participant E

Participants also said that the health coordination meetings were the best platform to provide direction for COVID-19 management as it supported coordinated response. One of the participants mentioned that the meetings were limited to information sharing only.

*"Umm. . . I want to reiterate the same thing. Like, if we look back at the situation during pandemic, the meeting worked as a platform in streamlining the necessary activities to be done. It provided us with direction. . ."*—Participant K

*"The health coordination meeting helped in information sharing. There was a good level of coordination among us. Therefore, I believe that health coordination meeting on basis of coordination was very fruitful."*—Participant G

*". . .I think it was more information sharing than coordination. In the beginning, it was okay, but later since its health cluster there required provincial level, other sub-clusters, it should have been more on coordination, policies, monitoring, I think we should have gone in that way, especially in the leading part, we should have more matured."*—Participant F

**Process.** Participants informed that the information was shared from the grass-root level initially, however, the information and communication were much more limited and repetitive later. The participants related the limitation in information to the decreased information from sub-clusters and grass-root level.

*"Yes, previously when cluster meeting started, provincial presentations were done, they advocated on the activities done in the districts and gaps assessment, but then the meeting went about 2 hours long, and so provincial health coordination meetings was formed. This resulted meeting to end in short time, but that formed gaps in updating activities done at field level, issues faced during the time of intervention were updated in live, which later diminished."*—Participant J

*"Lesson learnt shared during that time, in local level rather than thematic, practical way out was practiced. Activities carried out were addressed in the meetings before, but later it decreased, and I think we couldn't capture the information well."*—Participant G

*". . .certain monitoring of the sub clusters (Reproductive and Mental Health) could have been done because we lacked linkage and so whatever the sub clusters presented, we believed that to be right enough. Next, we should remember that we have fulfilled structure in the federal level, we fully utilized the district health office sources, during the earthquake and 2017 flood, the districts coordinated well during the time, later, the structure was also new and not all the provinces coordinated effectively."*—Participant C

*. . .we missed service delivery point from basic grassroot level like palikas, health coordination unit etc. So, what is the linkage point from them, either its PHEOCs or any other source, that kind of information flow was missing."*—Participant K

*"Next, in cluster coordination, it is not important for the same person or same division to share the information, of course, that's important but I think we should move much beyond that especially in a long row like 72 coordination meetings. . . The decrease in number of participants can indicate that our content and delivery was not appropriate enough. I felt like demand side from many local levels were missed, beneficiaries were never discussed. . ."*—Participant F

There were mixed responses on addressing issues and queries from the health coordination meetings. However, one of the participants admired the documentation of the meetings (meeting notes and documents).

*". . .the coordination meetings held before was enthusiastic and issues were addressed but during the second phase, our issues were not equally addressed, even in provincial level, they started shifting work to one another."*—Participant L

*"Also, usually meetings are one-sided but in health coordination meeting, questions in chat box were also addressed. This might seem minor, but in my point of view following the same trend over 72 health coordination meetings, this a very good initiative taken. . .The way that meeting notes are formalized with action points, this is very important in coordination. Likewise, this built confidence in us as well. Reference documents, guidelines are shared time and again during and after the meeting."*- Participant F

## Participation

One of the participants highlighted the need for the Terms of Reference (ToR) for all the stakeholders in the health cluster coordination mechanism.

*"During the beginning, health coordination was activated in a different manner from HEOC, ToR was not clear, some people even put forward their questions, another in logistics part, the information were confusing at times. Especially, during the staff turnover, many of us were unknown about our responsibility."*—Participant F

Sending of the link to the organization only and the conflicting schedules of the participants caused the missing of the meetings by some participants.

*"Our participation was frequent most of the times, but then there was constraint in sending meeting link. . .But some of them were missed due to conflicting schedules."*—Participant L

## Next steps

Participants were univocal on continuation of coordination meetings for preparedness, readiness, and response for disasters and other public health emergencies.

*"Similarly, health coordination should also focus not only on disaster, but also in post- disaster, like the COVID cases have been dropping, we can also focus on discussing the other important areas of health. Since monsoon is approaching, we can also decide if we can take up the issue in health coordination or start preparing now. Also, I think regular coordination needs to be continued. . .. I feel like we need to bring out updated health coordination partners in a loop and update in resource mapping. We should develop a resource sharing platform and*

*similarly, health coordination meetings can be regularized monthly. In general, health coordination mechanism needs to be in place.*"—Participant G

"*. . .Hence, this meeting should be regularized. Likewise, since monsoon is following, we*

*need to get prepared for the monsoon. Health coordination can look after the WASH and provincial clusters and update information. . .*"—Participant C

"*We can regularize health coordination and certain dedicated human resources should be in place. Nepal being challenged due to many aspects like climate, geography, etc., we cannot predict what comes in the way, information sharing mechanisms can be updated like via Viber group or any kind of emergency group.*"—Participant F

One of the participants also highlighted the role of health coordination meetings as the platform for progress monitoring and sharing.

"*. . .but another important work is to monitor, just to see what progress has been made and share the documents with all in a continuous process. It is also one of the critical roles of health coordination division, since it builds up the confidence for all the partners who actually are keen to work.*"—Participant F

## Suggestions/Recommendations

One of the participants suggested sharing experiences/expert views in the meetings and the connection with the provincial and local level to be maintained in the meeting.

"*There are a lot of places where we don't know exist, we need to seek expertise in those areas, the issues need to be addressed and existing guidelines should be easily accessible. The lessons learnt from our own experience need to be systematized in the upcoming days.*"—Participant E

"*Because federal ministry of health is to be responsible to coordinate in health emergency as being driven by constitution, so I think that message must be set up and due to work of, for example, health emergency response in the province and the local government are playing their part in health emergency plan or response plan. They should sanction coordination; they are willing to coordinate with ministry and related any support which*

is of ministry of health and ultimately health coordination cluster. . . Also, I think the

*connection between provincial and federal level should not be lost. We are discussing over here, but ground realities need to be addressed and solved, we need to consider those as well.*"—Participant E

Participants also requested to develop and/or revise the Terms of Reference (ToR) for Health Cluster Coordination Meetings.

"*Therefore, I think the ToR (Terms of Reference) of the health cluster must be reviewed in all federal, provincial and district level because we lack in the process of information flow system and the way forward can be reviewing of ToR and handover system should be improved so that we're on the same page now onwards. Even though the sub-clusters are not formed, we should keep going with the health cluster. . .. One of the action points could be that we can*

*include ToR from HEOC, and all the participants need to follow that. Agency's responsibility needs to be clear. . .. I think a revised ToR is a must. We should scale up the cluster activation like, during the normal period, we can get going with the preparedness while during emergency we get the activity done."*—Participant C

*"We need to identify the health partners and seek their own roles and responsibility in close coordination. Some relevant questions should be documented and addressed during or after the meeting."*—Participant G

*"During the beginning of cluster meeting, there were two kinds of institutions: 1) existing health institutions and 2) other non-health institutions that diverted their funds to work during COVID. Hence, we need to define their roles and update members so that they can engage beside COVID in later days. . .. Likewise, we do have a defined ToR for all the leads, co-leads, members etc. in our organization, maybe similar kind of defined ToR needs to be developed which might enhance effective participation further."*—Participant J

Participants requested for one door mechanism for support and information sharing, through the health coordination meeting.

*"But in context of coordination, two things seemed lacking, not only from center, but the entire system. Example, the needs/asks coming from different divisions, centers, was quite confusing. Rather than coming from multiple channels, it would have been quite easy if it carried one institutional value."*—Participant F

*". . .it should be improvised one door mechanism; there are different clusters, we have WASH cluster, and so many of the information were overlapping at times."*—Participant G

*". . .we created parallel system or multiple system, then there was struggle to share the information as quickly as possible. That was good. But I think we should have mainstreamed those multiple information systems, that created more confusion."*—Participant F

Participants suggested a mixed model for meetings with identification of focal person from each organization to reduce novelty of the meetings.

*"Hence, we can switch on to mixed module so that the issues, gaps, and challenges faced in field level can also be reached directly to federal level. Along with this, the meeting time can also be maintained, and it might be more effective that way. . .. Even when we observe in meeting, there are almost 30/40 organizations, that might not be able to physically attend meetings during other emergencies, hence, we can also opt for mixed model, i.e., virtual and physical so that there's limited physical presence."*—Participant J

*"Periodic sharing of template and updating it needs to be done and a focal person and alternative focal person needs to be allocated for information sharing to minimize gap.*

*These things need to be mandated."*—Participant C

*"I've regularly attended the HC meetings, as there are focal points in relevant clusters, cluster leads/co-leads/ any focal point need to be present. Because, we are discussing on the same topic, but not reaching conclusion."*—Participant J

*"Also, individual participation would be crowded, but allocating focal person would be much helpful."*—Participant L

*"We should have a defined resource and a limit to our responsibilities. Maybe, we can allocate a communication person, it is very important to have a dedicated person for information."*—Participant F

Participants also suggested developing a website for information sharing from and to the health cluster members.

*"Likewise, we can opt for some innovative ideas to address the issues, this might not require a long report but rather it'll be issue based, we can create a website for that, it'll more be like learning by doing."*—Participant E

*"It'll be better if we could develop a website for sharing that information. Forming an interactive dashboard and linkage of all partners could be effective for future reference as well."*—Participant F

## Discussions

This paper aimed to assess the performance of the Health Partner Coordination established for COVID-19 response in Nepal. The Ministry of Health and Population, Nepal activated the health coordination meetings early, based on the cluster approach as defined in the National Disaster Response Framework (2013) for COVID-19 response and conducted coordination meetings regularly to share information, identify gaps, and streamline efforts. Partners appreciated the leadership of MoHP to initiate these coordination meetings early, fixing the time for its conduction, and sharing information and ideas in the meeting. MoHP has experience of activating the cluster coordination mechanism and organizing the health cluster coordination meeting during the Nepal Earthquake 2015 [16], which might have supported the effective organization of these meetings during the COVID-19. The health coordination meetings were understood only as information sharing platforms by some partners and not as a platform for supporting coordinated response. Coordination meetings only conducted as a reaction to sudden disasters, rather than proactive meetings, have been identified as an area for improvement regarding coordination [17]. The meetings were more frequent during the initial phase of COVID-19 and during the surge of cases. It has been a common observation that almost all underdeveloped and developing countries prone to disasters respond to disasters in a reactive rather than in a proactive manner [18].

The meetings were documented with minutes shared to all the participants, which were praised by the health partners. However, it was found that in most meetings, follow-up through completion of action points were lacking and required improvement. As time went by, issues of repetition and monotonous content were observed, thus the suggestion to keep the meeting targeted to a specific objective and content relevant to the evolving situation. Coordination requires information scaled to different levels of responsibility and action and should be updated or revalidated as per the changing situations [19]. There were observations of limited information captured from the sub-clusters and ground level, yet if included this considerably increase the overall meeting duration. A balance must be sought to keep the meeting manageable and productive while noting that the lack of involvement of sub-national level is perceived as poor coordination, unclear lines of authority, and contributes to communication breakdown [20]. This may be addressed, in part, by having clear terms of reference and roles and responsibilities. Conflicting schedules and sometimes the unavailability of meeting details to the focal person were mentioned as some of the reasons for absence from meetings. At the same time, it was observed that having a fixed meeting schedule per week was

helpful for participants. Meetings with management issues can be labelled as a "bad" meeting [21], and thus require prior preparations and checks.

Using the health coordination meetings for sharing of experiences and expert views, involving, and reaching out to the provincial and local level, and one door mechanism for support and information sharing were some of the suggestions from partners. Terms of Reference for all the stakeholders in the health coordination mechanism were highlighted as the major need to ensure active participation. Development of the Terms of Reference (ToR) and a hybrid model (virtual and in-person) for meeting with identification of focal person from each organization were also suggested. Continuation of health partner coordination following the cluster approach for disasters, epidemics, and other public health emergencies and its use for progress monitoring and sharing was identified as way forward. Structural aspects of coordination bodies including availability of coordination structures and regular meeting fora; clear roles, mandates and sufficient authority; the value of building on existing coordination mechanisms; and ongoing functioning of coordination bodies, before and after crises are identified to be important for effective coordination [22]. Development of a website for information sharing from and to the health cluster members was also advised. A socio-technical decision support system has been identified to be a promising strategy for creating the workable balance between order and flexibility required for coordination in practice [19].

Two subfunctions of the health partner coordination were considered good by the health partners, one was recognized to be unsatisfactory, and the rest as satisfactory. Planning after identification of the priorities based on response analysis and acting in coordinated manner can support more effective response to the needs of the affected population. The purpose of the health cluster approach is to harmonize efforts and use available resources efficiently within the framework of agreed objectives, priorities, and strategies, for the benefit of the affected population(s) and towards collective outcome [23].

Data triangulation is a strength of this study that supported to gain a deeper and broader understanding of the health partner coordination, ensured objective analysis, and prevented researcher bias. The main limitation was retrospective data capturing which limited the scope of the data gathering based on an analytical framework.

## Conclusion

COVID-19 has highlighted the importance of effective coordination for response to pandemic. Nepal effectively faced the challenge of coordinated response to COVID-19 through early initiation of health coordination meetings and smooth functioning of the health coordination mechanism and processes. There were several recommendations like continuation of coordination meeting, conducting meetings in a hybrid model (virtual and in-person), development of ToR, regular progress monitoring and sharing, and development of website for information sharing to further strengthen the health coordination at different stages of a disaster or public health emergency response. The most repeated recommendation was the need for clarification of roles and responsibilities among humanitarian health partners made through the formulation of a clear terms of reference.

## Supporting information

**S1 File. Individual meeting's documents (Agenda, presentations, minutes) review tool.** (PDF)

**S2 File. Tool for assessment of health coordination established for COVID-19 pandemic in Nepal.**
(PDF)

**S1 Table. Interview guide for focus group discussion.**
(PDF)

**S1 Data. Datasets used in the research—compressed files archive.**
(7Z)

## Acknowledgments

We would like to express our sincere gratitude to Dr Roshan Pokharel, the secretary of the Ministry of Health and Population, and Dr Rajesh Sambhajirao Pandav, Country Representative for WHO Nepal for their overall leadership and support in developing this research manuscript, including in the organization of a manuscript writing workshop. We would also like to express our heartfelt gratitude to Melissa Kleine- Bingham for her review and editorial assistance to the final manuscript.

**Disclaimer**: The work represents the opinion of the authors and not that of the organization for whom they work.

## Author Contributions

**Conceptualization:** Sangeeta Kaushal Mishra, Samir Kumar Adhikari, Allison Eugenio Gocotano, Gaurav Devkota.

**Data curation:** Samir Kumar Adhikari, Pavan Kumar Sah, Subash Neupane, Barsha Thapa, Gaurav Devkota.

**Formal analysis:** Samir Kumar Adhikari, Pavan Kumar Sah, Subash Neupane, Barsha Thapa, Gaurav Devkota.

**Methodology:** Samir Kumar Adhikari, Allison Eugenio Gocotano, Gaurav Devkota.

**Software:** Gaurav Devkota.

**Supervision:** Sangeeta Kaushal Mishra, Allison Eugenio Gocotano.

**Writing – original draft:** Barsha Thapa, Gaurav Devkota.

**Writing – review & editing:** Sangeeta Kaushal Mishra, Samir Kumar Adhikari, Pavan Kumar Sah, Allison Eugenio Gocotano, Subash Neupane, Gaurav Devkota.

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
