## [Decision Letter · Decision Letter 0]

3 May 2024

PONE-D-24-09019Effectiveness of Health Partners Coordination for COVID-19 pandemic response in NepalPLOS ONE

Dear Dr. Devkota,

Thank you for submitting your manuscript to PLOS ONE. After careful consideration, we feel that it has merit but does not fully meet PLOS ONE’s publication criteria as it currently stands. Therefore, we invite you to submit a revised version of the manuscript that addresses the points raised during the review process.

Thank you for submitting your paper to the PLOS ONE journal. I enjoyed reading your paper which is interesting and important. I have provided comments and requires minor revisions. Please go through each comments in track changes version of the manuscript and address all comments. A final clean and track changes versions of the revised copies are required to upload once you complete.

Looking forward to receiving revised paper.

We look forward to receiving your revised manuscript.

With regards

Shalik Ram Dhital, PhD

Academic Editor

PLOS ONE

3. For studies involving third-party data, we encourage authors to share any data specific to their analyses that they can legally distribute. PLOS recognizes, however, that authors may be using third-party data they do not have the rights to share. When third-party data cannot be publicly shared, authors must provide all information necessary for interested researchers to apply to gain access to the data. (https://journals.plos.org/plosone/s/data-availability#loc-acceptable-data-access-restrictions)

a) A description of the data set and the third-party source

b) If applicable, verification of permission to use the data set

c) Confirmation of whether the authors received any special privileges in accessing the data that other researchers would not have

d) All necessary contact information others would need to apply to gain access to the data

4.Please review your reference list to ensure that it is complete and correct. If you have cited papers that have been retracted, please include the rationale for doing so in the manuscript text, or remove these references and replace them with relevant current references. Any changes to the reference list should be mentioned in the rebuttal letter that accompanies your revised manuscript. If you need to cite a retracted article, indicate the article’s retracted status in the References list and also include a citation and full reference for the retraction notice.

Reviewers' comments:

Reviewer's Responses to Questions

**Comments to the Author**

1. Is the manuscript technically sound, and do the data support the conclusions?

Reviewer #1: Yes

Reviewer #2: Yes

2. Has the statistical analysis been performed appropriately and rigorously? 

Reviewer #1: Yes

Reviewer #2: Yes

3. Have the authors made all data underlying the findings in their manuscript fully available?

Reviewer #1: Yes

Reviewer #2: Yes

4. Is the manuscript presented in an intelligible fashion and written in standard English?

Reviewer #1: Yes

Reviewer #2: Yes

5. Review Comments to the Author

Reviewer #1: Minor editorial comments the author needs to correct.

1 Methods section - Please replace conduction to conducting

2. Avoid starting a sentence with figure in the sentence "80.6% of the meetings covered all the agenda items while only ..........."

Reviewer #2: Dear Authors

I have reviewed the manuscript titled "Effectiveness of Health Partners Coordination for COVID-19 pandemic response in Nepal" submitted to PLOS One. Below is a summary of my assessment:

Originality and Significance:

The manuscript presents new insights into the effectiveness of health partner coordination for COVID-19 pandemic response in Nepal.

The research question is important and addresses a significant gap in knowledge regarding pandemic response strategies.

Methodology and Results:

The methodology is sound and appropriate for addressing the research question.

Results are logically presented and support the research findings.

Conclusion and References:

The conclusion is based on the research findings and appropriately drawn.

References provided are relevant and sufficient.

Language and Format:

There is room for improvement in the English language and adherence to PLOS One manuscript preparation guidelines.

I recommend revising the manuscript to enhance clarity and ensure compliance with formatting requirements.

Strengths and Limitations:

It is suggested to include a discussion of the strengths and limitations of the study to provide a comprehensive understanding of its implications.

Overall, the manuscript makes a valuable contribution to the field, but revisions are necessary to improve language quality and adherence to formatting guidelines. With these improvements, I believe the manuscript will be suitable for publication in PLOS One.

Thank you.

6. PLOS authors have the option to publish the peer review history of their article (what does this mean?). If published, this will include your full peer review and any attached files.

Reviewer #1: **Yes: **Prof. Tanimola Makanjuola AKANDE

Reviewer #2: **Yes: **Dr. Hari Prasad Kaphle, Associate Professor (Public Health), Pokhara University, Nepal

---

## [Author Response · Author response to Decision Letter 0]

31 Jul 2024

Dear Editor,

We were pleased to have an opportunity to revise our manuscript entitled “Effectiveness of Health Partners Coordination for COVID-19 pandemic response in Nepal”. In revised manuscript, we have carefully considered reviewers’ comments and suggestions. As instructed, we have attempted to succinctly explain changes made in reaction to all comments. We have replied to each comment in point-by-point fashion as well. We have also revised the manuscript text in track changes as the major ask was to make the language readable and understandable. The responses to the concerns raised by reviewers are provided below.

We would like to kindly note that the reviewers’ comments were very helpful overall, and we are appreciative of such constructive feedback on our original submission. After addressing the issues raised, we feel the quality of the paper is much improved.

Response to the comments from the reviewers:

Dear Reviewers, 

All the authors would like to thank you for your time to review the manuscript. We are highly obliged to the comments that have been provided in appreciation of the research in the subject matter of health partner coordination. Kindly find the responses to the comments provided for your perusal. 

Reviewer #1: Minor editorial comments the author needs to correct.

1. Methods section - Please replace conduction to conducting

Response: Respected Reviewer, thank you for noticing this. Use of the word “conduction” has been replaced within the manuscript.

2. Avoid starting a sentence with figure in the sentence "80.6% of the meetings covered all the agenda items while only ..........."

Response: Respected Reviewer, thank you for noting this. Sentences starting with figures have been managed as suggested.

Reviewer #2: Dear Authors

I have reviewed the manuscript titled "Effectiveness of Health Partners Coordination for COVID-19 pandemic response in Nepal" submitted to PLOS One. Below is a summary of my assessment:

Originality and Significance:

The manuscript presents new insights into the effectiveness of health partner coordination for COVID-19 pandemic response in Nepal.

The research question is important and addresses a significant gap in knowledge regarding pandemic response strategies.

Response: Respected Reviewer, thank you. The comments are well noted.

Methodology and Results:

The methodology is sound and appropriate for addressing the research question.

Results are logically presented and support the research findings.

Response: Respected Reviewer, thank you. The comments are well noted.

Conclusion and References:

The conclusion is based on the research findings and appropriately drawn.

References provided are relevant and sufficient.

Response: Respected Reviewer, thank you. The comments are well noted.

Language and Format:

There is room for improvement in the English language and adherence to PLOS One manuscript preparation guidelines.

I recommend revising the manuscript to enhance clarity and ensure compliance with formatting requirements.

Response: Respected Reviewer, we sincerely appreciate your insightful comments and suggestions for revising the paper. Necessary edits have been made within the revised manuscript as suggested.

Strengths and Limitations:

It is suggested to include a discussion of the strengths and limitations of the study to provide a comprehensive understanding of its implications.

Response: Respected Reviewer, we appreciate your insightful suggestion. Necessary inclusions have been made within the revised manuscript as suggested.

Overall, the manuscript makes a valuable contribution to the field, but revisions are necessary to improve language quality and adherence to formatting guidelines. With these improvements, I believe the manuscript will be suitable for publication in PLOS One.

Response: Respected Reviewer, we appreciate your insightful suggestions. Necessary inclusions have been made within the revised manuscript as suggested.

Dear Reviewers, kindly also find the response to the comments within the manuscript as necessary:

Please remove this short title as you have the similar title. Keep only full title.

Response: Respected reviewer, the comment is well noted. Since the ask for the short title was within the journal portal thus was included. It has been removed as suggested.

Keywords always write in small letters.

Response: Respected Reviewer, the comment is well noted and necessary edits made in the manuscript.

Please add Original reference here.

Response: Respected Reviewer, the comment is well noted and the necessary changes made.

Please fix your all reference with Vancouver style with Squared Bracket such as [1]. Please change through out the documents.

Response: Respected Reviewer, the comment is well noted and the necessary changes made throughout the document.

Please this reference

Sapkota K, Dangal G, Koirala M, Sapkota K, Poudel A, Dhital SR. Strategies for prevention and control of COVID-19 in Nepal. Journal of Patan Academy of Health Sciences. 2020 May 8;7(1):85-8.

Response: Respected reviewer, the comment is well noted and the suggested reference included.

Provincial level?

Response: Respected Reviewer, the comment is well noted. However, the term ‘subnational’ was the one used in the cited plan.

Could you please add one sentence here and say: Authentic communication was conducted by National Health Education, Information and Communication Center with in health and beyond, the public and private organizations including partners and local organizations. Add reference from annual report or NHEICC sources.

Response: Respected reviewer, the suggestion is well noted. Strategic preparedness and response plan for COVID-19 had many pillars of which coordination and communication were included in pillar one and pillar two respectively. As this research deals with only coordination aspect of COVID-19 strategic preparedness and response plan, thus communication has not been included in this research.

Moreover, during the activation of the Incident Command System of the Ministry of Health and Population, all the responses is being coordinated through the Health Emergency Operation Center, which also supported the daily media briefing and information sharing within health and beyond, in support to Health Coordination Division.

Do you means Ways?

Response: Respected Reviewer, the comment is well noted. The section is rewritten for better understanding, as suggested.

replace with 'conducting'

Response: Respected Reviewer, the comment is well noted and the necessary edit made.

This is better to start your results with 1-2 sentences then keep subheading.

Response: Respected Reviewer, the comment is well noted and the necessary additions made as suggested.

Avoid starting sentence s with figures

Response: Respected Reviewer, the comment is well noted and the necessary edits made throughout the document as suggested.

Please design this figure using PACE software and replace here. You can freely login PACE and upload your figure and create it.

Response: Respected Reviewer, the comment is well noted and the figure using PACE software provided.

Please prepare this table from PACE software and create new and upload.

Response: Respected Reviewer, the comment is well noted and the table using PACE software provided.

Please avoid italic

Response: Respected Reviewer, the comment is well noted and the necessary changes been made.

Create Table using Pace software.

Response: Respected Reviewer, the comment is well noted and the table using PACE software provided.

Avoid italic

Response: Respected Reviewer, the comment is well noted and the necessary changes been made.

Please start writing your discussion with the aims of this paper. Copy from the end of introduction and then further go with key findings.

Response: Respected Reviewer, the comment is well noted and the necessary additions made as suggested.

Please add one or two sentences here and say--

 Thiis process is almost similar with other meetings organizing and coordinating by MOHP and partners [Reference add]. This might be due to …….. Please why similarities? Give reason.. If your findings and previous results are different , give reasons of each finding. 

Response: Respected Reviewer, the comment is well noted and the necessary additions made.

Please add one word in Methods chapter explaining triangulation, If you say it is a strength. Such as Information was triangulated and verified during data analysis.

Response: Respected Reviewer, the comment is well noted and the and the necessary additions made as suggested.

Your ethical approval letter indicates that you spent NRS 605000.. But now, you said this research received no external funding. Contradiction?

Response: Respected Reviewer, the comment is well noted. The ethical approval required funding under different headings of the research, and the maximum budget from the NRs 605000 was estimated for the report writing and publication in a peer reviewed journal. 

This was spent for the cost of the NHRC ethical approval processing itself, and since PLoS did not require Article Processing Charge, the remaining amount is no longer used.

---

## [Editor Report · Decision Letter 1]

2 Aug 2024

Effectiveness of Health Partners Coordination for COVID-19 pandemic response in Nepal

PONE-D-24-09019R1

Dear Dr. Devkota

We’re pleased to inform you that your manuscript has been judged scientifically suitable for publication and will be formally accepted for publication once it meets all outstanding technical requirements.

Kind regards,

Shalik Ram Dhital, PhD

Academic Editor

PLOS ONE

---

## [Editor Report · Acceptance letter]

16 Aug 2024

PONE-D-24-09019R1 

PLOS ONE

Dear Dr. Devkota, 

I'm pleased to inform you that your manuscript has been deemed suitable for publication in PLOS ONE. Congratulations! Your manuscript is now being handed over to our production team.

Kind regards, 

on behalf of

Dr. Shalik Ram Dhital 

Academic Editor

PLOS ONE